# Exploration of Somatostatin Binding Mechanism to Somatostatin Receptor Subtype 4

**DOI:** 10.3390/ijms23136878

**Published:** 2022-06-21

**Authors:** Rita Börzsei, Balázs Zoltán Zsidó, Mónika Bálint, Zsuzsanna Helyes, Erika Pintér, Csaba Hetényi

**Affiliations:** 1Department of Pharmacology and Pharmacotherapy, Medical School, University of Pécs, 7624 Pécs, Hungary; rita.borzsei@gmail.com (R.B.); zsido.balazs@pte.hu (B.Z.Z.); monibalint18@gmail.com (M.B.); helyes.zsuzsanna@pte.hu (Z.H.); erika.pinter@aok.pte.hu (E.P.); 2János Szentágothai Research Centre & Centre for Neuroscience, University of Pécs, 7624 Pécs, Hungary; 3Algonist Gmbh, 1030 Vienna, Austria; 4PharmInVivo Ltd., 7624 Pécs, Hungary

**Keywords:** pocket, site, peptide, interaction, selectivity, dynamics

## Abstract

Somatostatin (also named as growth hormone-inhibiting hormone or somatotropin release-inhibiting factor) is a regulatory peptide important for the proper functioning of the endocrine system, local inflammatory reactions, mood and motor coordination, and behavioral responses to stress. Somatostatin exerts its effects via binding to G-protein-coupled somatostatin receptors of which the fourth subtype (SSTR4) is a particularly important receptor mediating analgesic, anti-inflammatory, and anti-depressant effects without endocrine actions. Thus, SSTR4 agonists are promising drug candidates. Although the knowledge of the atomic resolution-binding modes of SST would be essential for drug development, experimental elucidation of the structures of SSTR4 and its complexes is still awaiting. In the present study, structures of the somatostatin–SSTR4 complex were produced using an unbiased, blind docking approach. Beyond the static structures, the binding mechanism of SST was also elucidated in the explicit water molecular dynamics (MD) calculations, and key binding modes (external, intermediate, and internal) were distinguished. The most important residues on both receptor and SST sides were identified. An energetic comparison of SST binding to SSTR4 and 2 offered a residue-level explanation of receptor subtype selectivity. The calculated structures show good agreement with available experimental results and indicate that somatostatin binding is realized via prerequisite binding modes and an induced fit mechanism. The identified binding modes and the corresponding key residues provide useful information for future drug design targeting SSTR4.

## 1. Introduction

Somatostatin is a cyclic neuropeptide, widely expressed in both peripheral and central tissues. SST has two active forms, the 14 amino acid-long (referred to as SST throughout this study), and an *N*-terminally extended isoform of 28 amino acids [1,2,3,4]. Both forms are expressed in the same tissue areas, but it is not clear whether the same cells can produce them. SST is internally stabilized by a disulfide bridge between cysteine residues in positions 3 and 14 (Figure 1A).

SST inhibits the release of several endocrine hormones such as growth hormone, prolactin, thyrotropin, gastrin, insulin, secretin, and glucagon [3,5,6,7], and the local inflammatory reaction at the periphery [8,9]. As a neurotransmitter, SST plays role in many mechanisms centrally, such as pain transmission, mood coordination, and learning and behavioral responses to stress [3,10,11,12,13]. It has emerging therapeutic relevance for the diagnosis and/or the treatment of numerous diseases, such as type 2 diabetes mellitus, Cushing disease, Alzheimer’s disease, acromegaly, several neuroendocrine tumors, pain-associated conditions (inflammation, neuropathy, rheumatoid arthritis), and depression [6,14,15,16,17]. The native form of SST does not have clinical importance because of its short plasma half-life of 3 min [3] and various actions.

SST exerts its diverse biological effects via modulating somatostatin receptors (SSTRs). The therapeutic potential of SST–SSTR interactions is not fully utilized, and there is current pursuit for receptor-selective, orally-administrable drug candidates in several research groups and pharmaceutical companies [18,19,20,21,22,23,24,25]. SSTRs belong to the rhodopsin-like G-protein coupled receptor (GPCR) superfamily and contain seven transmembrane (TM) helices (Figure 1B) and extracellular (ECL) and intracellular (ICL) loops. ECL2 was suggested to play a major role in ligand binding and receptor activation. It was supported by mutational analysis and receptor chimera examinations, with the result that the ligand-binding pocket involves residues of TMs 3–7 and ECL2 that are responsible for high affinity ligand binding in all SSTR subtypes [26,27,28]. There are five SSTR subtypes named as SSTR1–SSTR5 with more than 50% sequence identity. The binding of SST is not SSTR subtype selective according to competitive radio-ligand measurements [29,30,31,32,33].

In this study, we focus on SSTR4 that proved to be a promising target in the treatment of inflammation and pain-associated conditions (neuropathic pain, neurogenic inflammation, bronchial asthma, rheumatoid arthritis), Alzheimer’s disease [34,35], and depression [36,37]. SST elicits anti-inflammatory and anti-nociceptive actions and can be released from the capsaicin-sensitive sensory nerve endings. This is mediated through the activation of SSTR4 [38]. Centrally, SSTR4 is involved in learning and memory processes [10] and anxiety and depression-like behavior [37]. Thus, SSTR4 agonists would be promising drug candidates with analgesic, anti-inflammatory, and anti-depressant actions. However, there is no potent SSTR4 selective, orally administrable drug on the market [39,40,41,42]. Several SST analogs are under development [43], and many of them are used in therapy, such as pasireotide, octreotide, dopastatin, lanreotide, or in diagnostics [44,45,46]. Most of the studies [31,47,48,49,50,51,52,53,54,55] investigated either the binding mode of several exogenous peptidergic SST analogs as drug candidates or the residues of SST taking part in ligand binding. There are only a few studies [26,56,57] that examined the binding properties of endogenous ligands to SSTRs, which might be explained with the lack of atomic resolution experimental structures of SSTRs.

The target-based rational design of new agonists necessitates the atomic resolution structure of SSTR4 and its complex with the native (endogenous) ligand SST. Indirect experimental [26,31,57,58] and theoretical [48,50,51,52,53,59] information has been accumulated on the approximate binding sites of SST on SSTR4. While the atomic resolution structures of subtype SSTR2 and its complexes were measured recently [60], experimental determination of the atomic resolution structure of SSTR4 has not been published.

In the present study, we investigate the binding mechanism of SST to SSTR4. Atomic resolution structures of the SST–SSTR4 complex are produced using an unbiased, blind docking approach, and the binding mechanism is explored using molecular dynamics simulations in an explicit water model. We investigate if SST follows a “lock and key” or rather an induced fit mechanism and if it adopts prerequisite binding modes while approaching the final binding pocket on SSTR4.

## 2. Results and Discussion

### 2.1. The Structure of SSTR4

The experimental determination of the atomic resolution structure of SSTR4 has not been accomplished yet (Introduction). Homology modelling is an alternative method of choice [40,41,48,51,52,53,61,62] for producing SSTR structures. Building a good SSTR model necessitates the selection of a template protein of good sequential agreement with the receptor. The first homology modeling study of SSTR4 [51] used the active form of the β_2_ adrenergic receptor (PDB code: 3p0g) as a structural template. In recent years, new template structures have emerged, and our BLAST [63] (Methods) search resulted in a list of new template proteins (Appendix A) in the Protein Databank (PDB, [64]). A comparison of the homology models built from the templates led to the selection of the active form of the μ-opioid receptor (PDB code 5c1m) as a new template of SSTR4, also used in a previous study as a template of receptor subtypes SSTR2 [48]. The homology models generated from the old (3p0g) and new (5c1m, Figure 1B) templates showed overall similarity (Appendix A) in the position of TM helices and differences in ECL2 possibly involved in ligand binding.

### 2.2. The External Binding Mode

Following generation of the homology model of SSTR4, a modified fragment blind docking (FBD) approach [65,66] was applied to locate the binding pocket of SST targeting the entire surface of receptor. The approach allows an unbiased (blind) detection of anchoring points of SST without prior information on the location of the binding pocket. Structure–activity relationship studies have shown [31,33,55,67] that central amino acids F7W8K9T10 (Figure 1A) play a pivotal role in SST binding activation of the SSTRs. The disulfide bond between C3 and C14 (Figure 1A) largely determines the positioning of FWKT in the apical β-turn region of the SST structure [31,68,69,70]. Accordingly, this FWKT fragment was used as a seed during FBD to locate the binding mode of the central region of SST on SSTR4. The entire surface of the (3p0g-based) homology model of SSTR4 was covered by a mono-layer of the copies of the blocked tetrapeptide fragment (Ace–FWKT–NHMe) using several wrapping cycles ([65] Section 3). After seven cycles, 74 copies covered the entire surface of the SSTR4 target (Figure 1C). The docked binding mode of the best interaction energy (E_inter_, Appendix A) found an extracellular binding cleft formed by the ECL1–3 (Figure 1C) regions of SSTR4.

The tetrapeptide–SSTR4 complex structure was used to construct the full length SST molecule in the binding cleft. This was achieved by a somewhat unusual application of the popular homology modelling program Modeller [71]. The program was instructed to grow the remaining ten amino acids of SST (Methods) in the binding cleft of the SSTR4 target structure (Methods), extending the tetrapeptide seed (Figure 2A). The resulting SST (full length)–SSTR4 complexes were energy-minimized, and the corresponding interaction energies were calculated (Methods). The SST structure in the raw complex with the best E_inter_ after the growing step (Figure 2B) did not adopt the above-mentioned β-turn structure at the FWKT region and resided at the extracellular surface of SSTR4 (see Section 3 for identifying criteria of a β-turn structure). SST also did not form a salt bridge with D126, a key residue involved in SSTR4 activation [26,28,56,57,58,70,72]. D126 is located deeper in the transmembrane region of TM3 (Figure 1B) and expectedly formed a salt bridge with the apical K9 in the final binding mode of SST.

Due to the apparent disagreement of the raw SST–SSTR4 complex with the above-mentioned literature data, it was subjected to further refinement in a 350 ns-long MD simulation (Section 3). The expected [31,33,55,67] β-turn structure of SST appeared for longer periods during the MD simulation. Migration of SST was also observed towards the transmembrane region, as indicated by the slight decrease of the distance of the expected SST:K9-SSTR4:D126 salt bridge (d_SB_) from 21 (Figure 2B) to 18.5 Å (Figure 2C). In the MD-refined structure, the interaction of SST:K9 with ECL3 was broken down, while the connection of the tail regions of SST with ECL2 and ECL3 remained (Figure 2C,D).

During the MD refinement, movement H-bonds of SST:K9 with the target residues on ECL3 were broken down, and instead of the apical K9, backbone oxo groups of SST formed anchoring salt bridges with positively charged amino acids (R188, R191) of ECL2 (Figure 2D, Appendix A). Interactions between SST and ECL2 were reasonable, as ECL2 is known [27] to have a lid function in SST association. Thus, the external binding mode identified at the ECL2 lid (Figure 2C,D) is certainly a prerequisite state en route to the internal binding mode.

### 2.3. The Internal Binding Mode

To construct the final, internal binding mode, the SSTR4–tetrapeptide complex (Figure 2A) was subjected next to a 100 ns-long MD simulation, where the tetrapeptide and the ECLs moved freely, but position restraints were applied on the TMs (Methods). It was expected that the tetrapeptide would find the internal binding mode faster than the full length SST due to its higher translational and conformational mobility. As can be seen in Figure 3A, d_SB_ decreased from the initial 18.5 Å to about 10 Å (red squares in Figure 3A) several times, which may indicate the presence of a stable intermediate conformation of SST between its external and internal binding modes. From the 83rd ns, the fluctuation of d_SB_ decreased, reaching the lowest distance (5.2 Å) by 98.2 ns.

Similarly to the previous section, the full length SST molecule was grown from amino acid K9 (of aFWK9Tm) as a seed, with the lowest d_SB_ of 5.2 Å observed during MD (Figure 4A). The growing process (described in detail in Methods) resulted in three full length SST–SSTR4 complex structures subjected to three, respective, 350 ns-long MD simulations. In two of the MD simulations, SST:K9 reached a d_SB_ of 3.1 Å (Figure 3B and Figure 4B) and 3.0 Å (Figure 3B and Figure 4B), respectively. The third MD resulted in a backward movement of SST towards the external binding mode (increasing d_SB_ in Figure 3D). The interaction patterns (Appendix A) of the internal binding modes described in Figure 4B,C were similar, and the one with a d_SB_ of 3.0 Å was selected as an internal binding mode for further description. In the internal binding mode, the position of SST was stabilized by salt bridges, and H-bonds formed with SSTR4 residues, including D126, N199, D289, and Y301 (Figure 4D).

### 2.4. The Binding Mechanism

The MD simulations of the previous section shed light on the association of SST with SSTR4 and its movement back to the external binding mode. Both associative MDs indicated that there were two highly occupied intermediate binding modes at a d_SB_ of 5–6 Å and 10 Å (Figure 3B,C), respectively. Notably, the intermediate at 10 Å was also identified in the simulation of the tetrapeptide–SSTR4 complex (Figure 3A). The steps of the associative movements were visualized (Figure 5, Appendix A) and showed a considerable conformational change of SST during the binding process. The conformational flexibility of SST was the most pronounced at its apical region, which showed a large flip between the internal and external binding modes (Figure 6A).

Similarly, SSTR4 also underwent a conformational change when SST moved from the intermediate state (d_SB_ = 10 Å) to the external binding position. The gap formed by ECL2 and ECL3 of SSTR4 increased to let the ligand dissociate from the receptor (Figure 3D and Figure 6B). In agreement with our findings, oligopeptides such as SST are known to activate their receptor via an induced fit mechanism very common in similar receptor activation processes involving considerable conformational changes on both the target [73,74,75] and the ligand [74,76,77] sides.

Furthermore, the intermediate state at d_SB_ = 5–6 Å (Figure 5) was stabilized by a network of water molecules in the interface and linked the apical region of SST to SSTR4, as shown in a close-up (Figure 6C). There were three water molecules connecting D126, Y301, and SST:K9 via a H-bonding network (Figure 6C). The role of such networks has been described by recent studies [78]. However, the internal binding mode was finally stabilized only by the SST:K9-SSTR4:D126 salt bridge (d_SB_ = 3.0 Å) without the above interfacial water molecules (Figure 6D), indicating that a de-hydration process took place in the final binding step. Several target amino acids (A197, C198, N199, and D289) were involved in both the external and internal binding modes. These residues assist the transition movement of SST from the external towards the internal binding mode (see also Appendix A).

All-in-all, the binding mechanism of SST to SSTR4 involves a migration between external and internal binding modes via intermediate states stabilized by water networks. The binding involves a conformational flip in the apical β-turn region of SST.

### 2.5. Comparison of SST Subtype Binding

Recent determination of the atomic resolution structure of the SSTR2–SST complex [60] allowed for comparison of the binding modes of SST on SSTR2 and SSTR4. The internal binding mode of SST on SSTR4 (Section 4) was used for this comparison. A per-residue energy analysis of the SSTR-SST interaction energy (E_inter_) showed that residues D122(D126), S279(S287), Y302(Y301) are important for binding of SST to both SSTR2(SSTR4) receptor subtypes (Figure 7). D122 proved to be essential in receptor activation [56,57,72]. The E_inter_ pattern on the SST side (Figure 6B) showed that residues A1, G2, K4, K9, and C14 are important in the interaction with both receptor subtypes. A difference could be observed at N5 and F6 (I284, V280, Y205, E200, R184) positions, preferring SSTR2, while F7 and F11 (D289, T286, L200, N199) are involved in the SSTR4 complex. The role of Fs and K4 was also suggested by previous alanine scanning studies [29].

An overall ca. 180° flip of the binding conformation of SST (Appendix A) could be observed between the internal binding mode on SSTR4 if compared with that of SSTR2 [60]. An E_inter_ analysis was also performed for the alternative binding mode of SST (observed in [60]) on SSTR4 (see Methods for details of construction of the complex). The E_inter_ plots (Appendix A) showed that K4, K9, and C14 (SST) and D126, S287, and Y301 (SSTR4) are important in all binding modes. F6 has importance only in case of SSTR2. W8 and N5 (on the SST side) and L283 and Q201 (on the receptor side) were identified as important residues only in the alternative binding mode. The above differences in SST binding to SSTR2 and 4 may serve as a good starting point in the design of subtype-selective SST analogues.

## 3. Methods

### 3.1. The Structure of SSTR4

A BLAST (Basic Local Alignment Search Tool) [63] search with Blosum 62 substitution matrix using a conditional compositional score matrix adjustment at NBCI [79] against the PDB Database [64] was applied to identify the template candidates for model building. The BLAST search resulted in 100 PDB codes. They were ranked according to their total scores. The best ranked template candidates were 4n6h, 4rwa, 6dde, and 5c1m (Appendix A). The structure of the δ-opioid receptor bound to a bifunctional peptide (PDB code: 4rwa) was excluded. Structures 5c1m and 6dde represent the crystal structures of agonist binding μ-opioid receptors, and 5c1m had a better resolution (2.1 Å compared to 3.5 Å for 6dde). The A chain of both the human δ-opioid receptor (4n6h) and the active form of the μ-opioid receptor (5c1m), and, furthermore, the active form of the β_2_-adrenergic receptor (3p0g) used in a previous study were employed for model building described in the paper of Liu et al. [51]. SSR4 sequence was taken from the UniProt database (P31391 (37-330)) the not-aligned *N* and *C* terminals were cut). After the sequence alignment using the Modeller program package [71], ten models were generated from each template, and models with the lowest Discrete Optimized Protein Energy (DOPE) score were further investigated (Appendix A). The RSMD value of CA atoms for the best models was calculated (Appendix A). The models were superimposed, and their structures were compared. Due to the high similarity of the opioid receptor-derived models, only the μ-opioid receptor (5c1m)- and β-adrenergic receptor (3p0g)-based homology models were used for further investigations.

### 3.2. The External Binding Mode

#### 3.2.1. Fragment of SST

The NMR structure of SST dissolved in 5% D-mannitol is known (PDB code: 2mi1). The apical region of SST, F7-W8-K9-T10, was extracted, and its *N* and *C* terminals were capped with acetyl and *N*-methyl groups (Ace–FWKT–NHMe) to neutralize the terminal charges. This Ace–FWKT–NHMe was used for docking calculations.

#### 3.2.2. Energy Minimization

A uniform two-step energy minimization process in AMBER99SB-ILDN force field by GROMACS [80] was used prior to MD simulations. Molecules were placed in the center of a cubic box with the distance of 10 Å between the box and the solute atoms. The simulation box was filled with TIP3P explicit [81] water molecules and counter ions to neutralize the total charge of the system. The convergence thresholds of the first (steepest descent) and second (conjugant gradient) steps of minimization were set to 100 and 10 kJ mol^−1^ nm^−2^, respectively.

#### 3.2.3. Docking Calculations

The energy-minimized target structures were used in docking calculations. The Wrapper module of the WnS method [65] was applied for Fragment Blind Docking (FBD) during which the entire surface of the target (3p0g) was covered by a mono-layer of the Ace–FWKT–NHMe copies by a series of blind docking cycles performed by AutoDock and AutoGrid [82]. Docking parameters were used for FBD, as described in our previous studies [65,83]. Wrapping the target into ligand copies allows systematic mapping of all possible binding modes of a ligand. At the end of wrapping, the fragment bound with the lowest E_inter_ was chosen as the best ligand position (Appendix A). The resulting docked complex was superimposed on the receptor structure of 5c1m and used in the next growing step. The distance between the amino *N* atom of K9:SST and the carboxylate *C* atom of D126:SSTR4 (d_SB_) was determined. The docking calculations were not focused on a selected region of the protein, and the ligand could navigate without positional or torsional constraints during docking. Thus, the blind docking calculations were unbiased without the use of previous knowledge of the binding site. The binding modes of the ligand covered the entire surface of the protein after blind docking with Wrapper (Figure 1C). The binding mode with the most favorable calculated E_inter_ was selected for further homology modelling steps.

#### 3.2.4. Growing of SST into the Binding cleft

The full-length ligand was built into the receptor using the fragment as a seed by the homology modelling approach. SSTR4-FWKT (Ace–FWKT–NHMe without the capping groups) structures were used as templates, and the query sequence was the sequence of the receptor and the full length ligand together taken from UniProt database (SSTR4: P31391 (37–330), like the homology models, SST: P61278 (103–116)). Structure alignment was manually optimized to obtain the identical regions correctly under each other. The Modeller [71] program package was applied to build ten models for each template. Explicit manual restraint was added to generate the disulfide bond in SST. As the DOPE score was very similar for all generated models, the E_inter_ (Appendix A) values were calculated [84] (Lennard–Jones energy, Amber parameters [85,86]) and applied for model selection.

#### 3.2.5. Molecular Dynamics Simulation

For identifying the internal binding mode of SST and investigating its binding mechanism on SSTR4, a series of MD simulations was applied in the TIP3P explicit water model with the AMBER99SB-ILDN force field using the GROMACS program package following two step energy-minimization (described in Section 3.2.2). In all cases, the target was treated as a rigid body, except the ECL regions (37–42; 109–225; 184–208; 284–294), to allow the entrance of the ligand into the receptor. Position restraints were applied on the heavy atoms of TMs with a force constant of 100 kJ/mol^−1^ nm^−2^. For temperature coupling, the velocity rescale and the Parrinello–Rahman algorithm were used. Solute and solvent were coupled separately with a reference temperature of 310.15 K and a coupling time constant of 0.1 ps. The protonation states of amino acids were set according to pH 7.4. Pressure was coupled by the Parrinello–Rahman algorithm and a coupling time constant of 0.5 ps, compressibility of 4.5 × 10^−5^ bar^−1^, and reference pressure of 1 bar. Particle Mesh–Ewald summation was used for long range electrostatics. Van der Waals and Coulomb interactions had a cut-off at 11 Å. Periodic boundary conditions were treated after the finish of the calculations. After each trajectory, the periodic boundary effects were handled, the system was centered in the box, and target molecules in subsequent frames were fit on the top of the first frame. The final trajectory including all atomic coordinates of all frames were converted to portable xdr-based xtc binary files.

#### 3.2.6. MD Refinement in the External Binding Cleft

The SSTR4–SST complex was submitted to a 350 ns-long MD simulation described above, and d_SB_ was calculated throughout the MD simulation using the gmx distance modul of GROMACS. The structure with the smallest d_SB_ was determined as the external binding mode of SST on SSTR4. Interacting target residues within a 3.5 Å distance of SST were determined and listed (Appendix A).

### 3.3. The Internal Binding Mode

#### 3.3.1. Molecular Simulation for Exploring the Internal Binding Cleft

The exploration of the internal binding mode of SST was performed similarly to the external one; however, the location/position of the SST tetrapeptide seed was determined by a 100 ns-long MD simulation instead of docking. After the two-step energy minimization process, the SSTR4–Ace–FWKT–NHMe complex (5c1m-based model with the superimposed aFWKTm) was submitted to a 100 ns-long MD simulation.

#### 3.3.2. Growing the Full Length SST into the Receptor

After the 100 ns-long MD, the structure with the smallest d_SB_ was used to build the full length SST into the receptor similarly to the external binding mode. After generating the homologies (similar to the method of external binding cleft), many close contacts occurred in the structures that remained also after the two step energy minimization procedure. Thus, in this case, instead of the whole apical FWKT region, only the K9:SST was used as a “seed” for building the ligand. Models (3 × 10) were generated using no, 5 Å, and 6 Å distance restraints on d_SB_, respectively, and in all homologies, d_SB_ was determined again (Appendix A). Models with the smallest d_SB_ from each group were further investigated using MD simulation.

#### 3.3.3. MD Refinement in the Internal Binding Cleft

The energy-minimized models with the smallest d_SB_ distance from each group (Appendix A) were submitted to a separate 350 ns-long MD simulation to investigate the associative and dissociative movements of the ligand. Calculation of d_SB_ was performed throughout each MD simulation, and structures having the smallest one were determined as the internal binding position of SST, and the interacting target residues within 3.5 Å distance from the ligand were determined (Appendix A).

### 3.4. Comparison of SST Subtype Binding

#### 3.4.1. Determination of Interacting Energy per Residues in SSTR4/SSTR2–SST Complexes

Following the two step energy minimization procedure, Coulomb intermolecular interaction energies were calculated [84] with a distance-dependent dielectric function [87] and Amber partial charges [85,86] globally and per residues for both SSTR2–SST and internal SSTR4–SST complexes. Comparison of the per residue interacting energies was based on the sequence alignment of the targets created by EMBOSS Needle [88].

#### 3.4.2. Energy Analysis of Alternative Binding Mode

There was a ca. 180° flip of the binding conformation of SST (Appendix A) in the internal binding mode on SSTR4 compared with that of SSTR2. Thus, the SSTR4–SST complex with this alternative binding mode was constructed by superimposing the targets. Following a two-step energy-minimization, a global and per residue E_inter_ analysis was also performed for this structure.

## 4. Conclusions

The present study investigated the binding mechanism of SST to SSTR4. While the SST–SSTR2 structure was recently published, the atomic level complex of SST and SSTR4 has not been determined yet. As SSTR4 also plays an important role in the pathobiochemistry of various diseases (Introduction), we thus focused on the calculation of SST–SSTR4 complex structures. Beyond the complex structures, the dynamics of the binding mechanism of SST was also elucidated, and key binding modes (external, intermediate, and internal) were distinguished. The role of induced fit and hydration was discussed. The most important residues on both receptor and SST sides were identified. Finally, an energetic comparison of SST binding to SSTR2 and 4 offered a residue-level explanation of receptor subtype selectivity. In good agreement with experimental results, we found that the extracellular regions of helices and loops play an important role in SST binding, and structural differences in these regions are important in receptor subtype selectivity. The detailed structural comparison of SST binding to SSTR2 and 4 helps in the development of new, subtype, and disease-selective SST analogues.

## Figures and Tables

**Figure 1 ijms-23-06878-f001:**
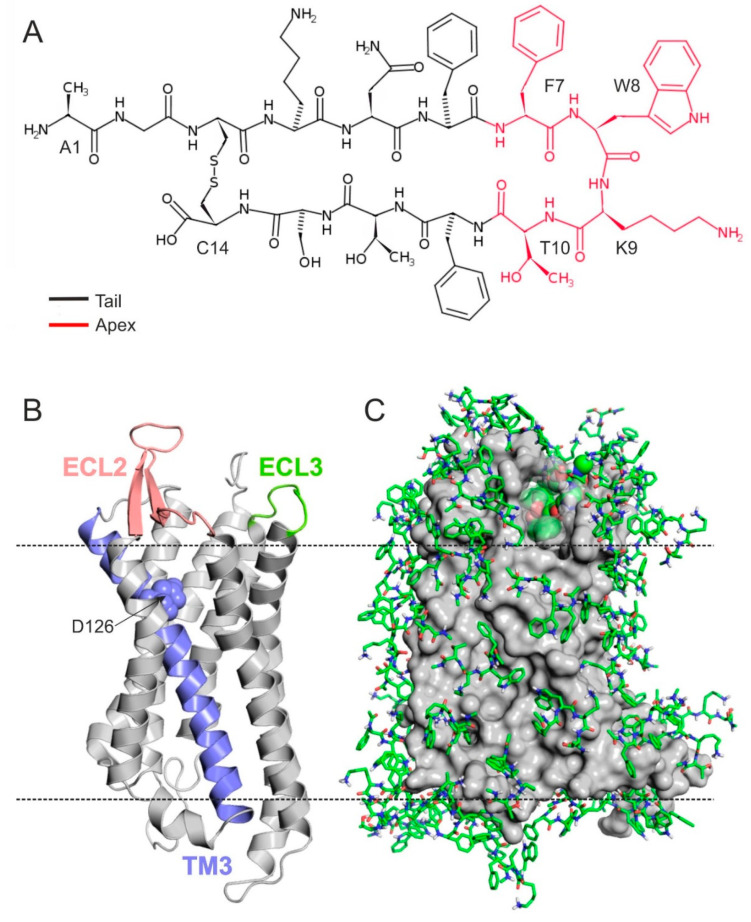
(**A**) Lewis structure of SST highlighting the apical FWKT region with red. (**B**) Homology model of SSTR4 in cartoon representation. D126 (spheres) on TM3 (teal), ECL2 (salmon), and ECL3 (green) are proved to be important in ligand binding and receptor activation. (**C**) SSTR4 (grey, surface) covered with monolayer of tetrapeptide fragment (Ace–FWKT–NHMe) copies (green, all atom, sticks) at the end of the 7th docking cycle. The best energy fragment is highlighted with spheres (green, all atom).

**Figure 2 ijms-23-06878-f002:**
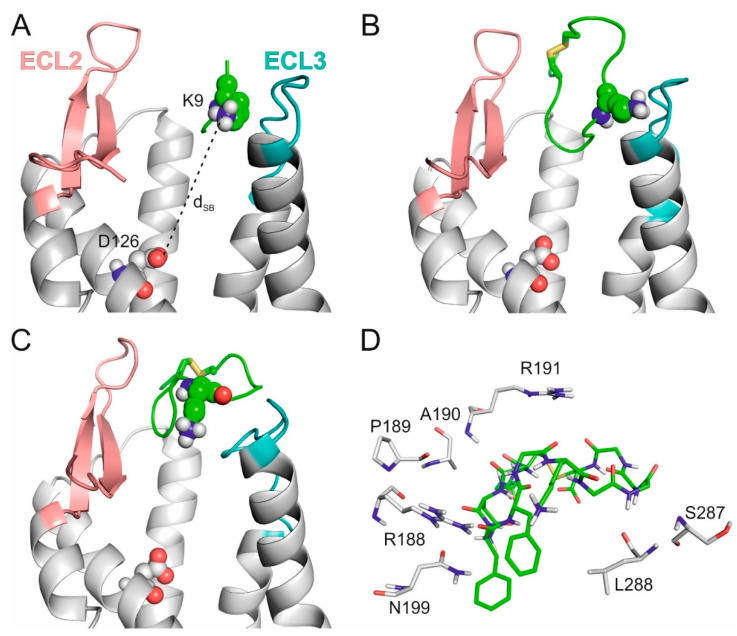
(**A**) SSTR4 with the best energy tetrapeptide fragment at the end of WNS. (**B**) The energy-minimized SSTR4–SST complex built from the “seed” of the fragment by homology modelling. (**C**) The SSTR4–SST complex in the prerequisite external binding mode after MD refinement. In (a, b, and c) ECL2 (salmon), ECL3 (teal), and D126 (spheres) are highlighted on SSTR4 (grey, cartoon) K9 of SST, and its fragment (green, cartoon) is in spheres representation. (**D**) The close-up view of the external binding mode of SST (green, sticks, all atom) with the target residues (grey, sticks, all atom) being within 3.5 Å distance of the ligand.

**Figure 3 ijms-23-06878-f003:**
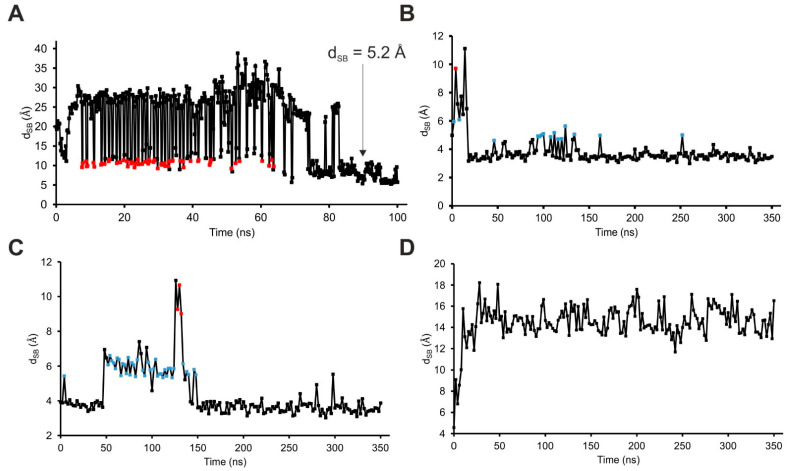
(**A**) D_SB_ plot of SSTR4–tetrapeptide fragment complex MD simulation for exploring the internal binding cleft of SST; (**B**–**D**) D_SB_ plots of the MD refinement of the three SSTR4–SST models containing the ligand in the internal binding cleft. In two of these simulations (**B**,**C**), SST was able to create a salt bride with D126 (d_SB_ = 3.1 Å and 3.0 Å), but in the third one (**D**), the dissociation of SST could be observed. Intermediate states are colored with red (d_SB_ = ~10 Å) and blue (d_SB_ = ~5 Å) points.

**Figure 4 ijms-23-06878-f004:**
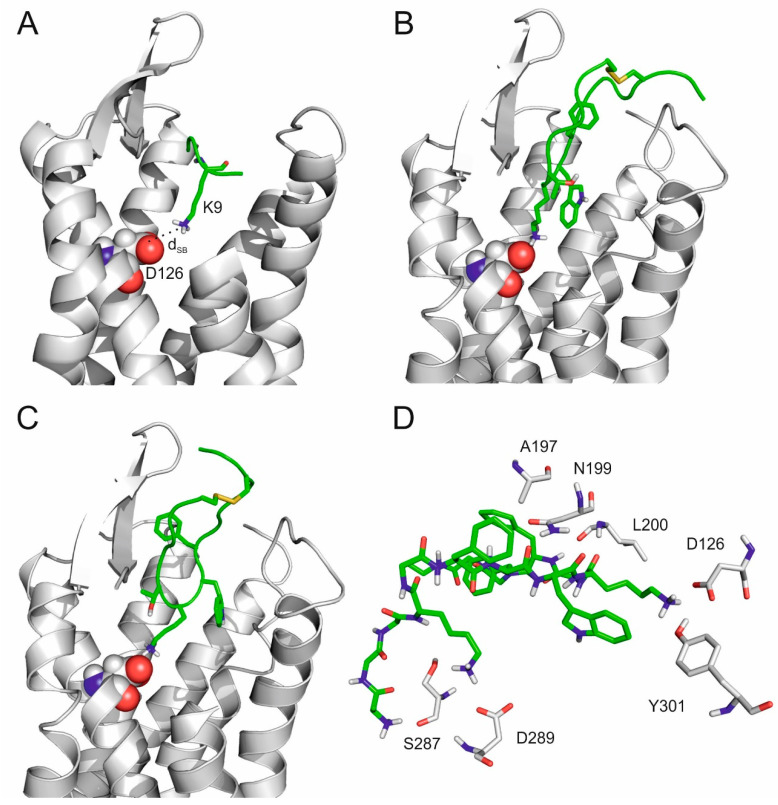
(**A**) Complex of SSTR4 (grey, cartoon, D126 highlighted by spheres) and the tetrapeptide fragment (green, K9 highlighted by sticks) with the smallest d_SB_ in the 100 ns-long MD simulation. (**B**,**C**) Internal binding mode of SST with 3.1 Å (**B**) and 3.0 Å d_SB_ determined in separate 350 ns-long MD simulations. (**D**) The close-up view of SST (green, sticks) in the internal binding mode surrounded with target residues (grey, sticks) within a 3.5 Å distance from the ligand.

**Figure 5 ijms-23-06878-f005:**
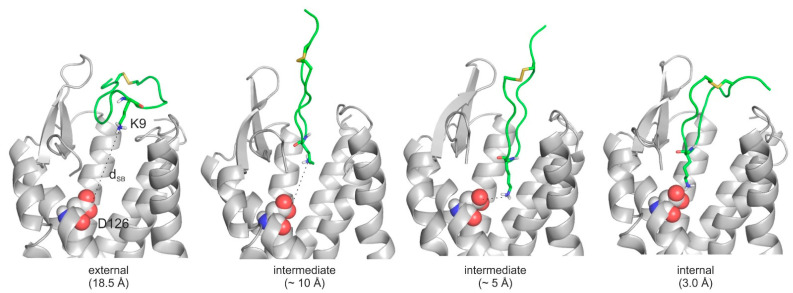
Main steps of the binding mechanism of SST (green, cartoon, K9 highlighted with sticks, all atoms) including external, intermediate (~10 Å and ~5 Å), and internal binding modes on SSTR4 (grey, cartoon). D126 is highlighted with spheres. This binding mechanism is illustrated in Appendix A.

**Figure 6 ijms-23-06878-f006:**
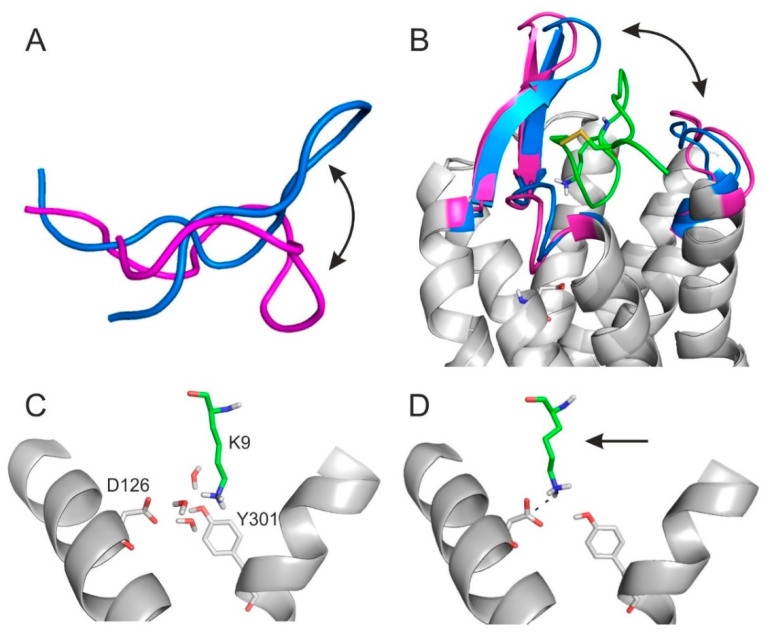
(**A**) The conformational change of SST during binding to SSTR4: internal (marine) and external (magenta) binding conformation of SST (cartoon) aligned by their tail regions. (**B**) Opening (magenta) and closing (marine) movements of the lid including ECL2 and ECL3 during the binding and dissociation of SST (green, cartoon, K9 highlighted with sticks, all atom). (**C**) The close-up view of the intermediate state (~5 Å) (Figure 5). The three water (grey, sticks, all atom) molecules help the connection of SST and SSTR4. (**D**) The close-up view of the final internal binding mode of SST (Figure 5) on SSTR4 after dehydration and movement (arrow) of SST:K9. In (**C**,**D**) K9:SST and SSTR4 are in green, sticks, all atom and grey, cartoon, D126, Y301 highlighted with sticks, all atom representation, respectively.

**Figure 7 ijms-23-06878-f007:**
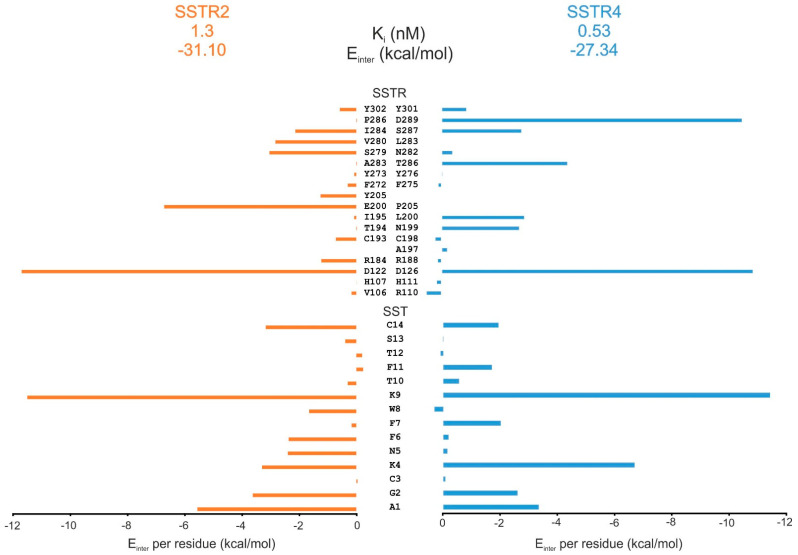
Per-residue E_inter_ contributions of SSTR2–SST (orange) and SSTR4–SST (blue) complexes shown for both the receptor (SSTR) and the somatostatin (SST) sides. Note that different amino acids may appear at identical positions in the sequences of SSTR2 and 4 after sequence alignment, as listed in the SSTR-based analysis (top). Source of K_i_ values is [32].

## Data Availability

Not applicable.

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
