# Peer review of "Exploration of Somatostatin Binding Mechanism to Somatostatin Receptor Subtype 4"

_ijms, 2022, doi:10.3390/ijms23136878_

Round 1

Reviewer 1 Report

Dear authors,

A very well presented work with a lot of important findings concerning the SSTR4. There are only two remarks I would like to highlight:

1) What kind of bias you refer too on your blind docking approach? you mean inserting a constrain or focusing to just one region would be biased? please elaborate a bit more in the text to be more specific

2) Since the SST-SSTR2 complex is now available over the protein data-bank it would be interesting to provide the RMSD between your model and the complex, adding also a couple of lines analyzing SST binding mode and similarity witnessed in your case.

Author Response

Reviewer 1

A very well presented work with a lot of important findings concerning the SSTR4. There are only two remarks I would like to highlight:

1) What kind of bias you refer too on your blind docking approach? you mean inserting a constrain or focusing to just one region would be biased? please elaborate a bit more in the text to be more specific

The Reviewer is acknowledged for his/her careful work and suggestions. Answering the above questions, the following text was inserted (lines 313-319): „The docking calculations were not focused on a selected region of the protein, and the ligand could navigate without positional and torsional constraints during docking. Thus, the blind docking calculations were unbiased without the use of previous knowledge of the binding site. The binding modes of the ligand covered the entire surface of the protein after blind docking with Wrapper (Figure 1C). The binding mode with the most favorable calculated Einter was selected for further homology modelling steps.”

2) Since the SST-SSTR2 complex is now available over the protein data-bank it would be interesting to provide the RMSD between your model and the complex, adding also a couple of lines analyzing SST binding mode and similarity witnessed in your case.

An RMSD of 1.538 Å can be measured between the Cα atoms of the experimental SSTR2 structure (PDB code 7t10) and our SSTR4 model (internal state). Notably, SSTR2 and SSTR4 belongs to different receptor sub-type families (with several sequential differences), and therefore, a full structural agreement cannot be expected. However, the RMSD value reflects an overall match between the backbone parts (mostly at the TM helices). Although it was not aimed in the study, we also performed the homology modeling of SSTR2 using the same template (µ-opioid receptor, PDB code 5c1m) and method and compared to the measured SSTR2 (PDB code 7t10). Here, an overall match with RMSD values of 1.306 â„« and 1.237 â„« were obtained for the full protein and the binding region (in a 3.5 â„« distance around SST), respectively. As it was discussed in the main text, structural differences between SSTR2 and SSTR4 were expected and found in the extracellular regions including the C-terminal of TM4, ECL2, C-terminal of TM6, ECL3 and N-terminal of TM7 which showed somewhat larger structural deviation. These regions (especially ECL2) take place in ligand binding, and suggested to play important role in receptor selectivity. The latter finding is summarized in the Conclusions (lines 402-406).

Reviewer 2 Report

The manuscript entitled “Exploration of somatostatin binding mechanism to somatostatin receptor subtype 4“ by Börzsei et al. investigates the chemical basics of endogenous ligand binding to SSTR4. The study was conducted by an in silico approach using BLAST search to find the adequate model of SSTR4 and investigate its interaction with SST. The study seems to be appropriately designed and carefully performed. There are some points that I would like to address:

Introduction:

ll. 46-48: “As a neurotransmitter SST plays role in lots of mechanisms centrally like pain transmission, motor and mood coordination, learning and behavioral responses to stress, locomotor activity [3,10–13]“. I have not yet come across a study showing that SST influences motor coordination and/or locomotor activity. To my knowledge, there is no original article showing that SST influences locomotor activity and motor coordination (motor coordination and locomotor activity does not refer to GI tract motility). The articles cited by the authors do not support that statement.

ll. 69-71: Please clarify the first part of this sentence, the statement of the first main clause is not entirely clear. “SSTR4 mediates anti-inflammatory and anti-nociceptive actions of SST without any endocrine effects releasing from the capsaicin sensitive sensory nerve endings at the periphery [38], and centrally it is involved in learning and memory processes [10] and anxiety and depression-like behavior [37].

 Please carefully check the spelling of SSTR4. Somatostatin receptor 4 for example seems to be abbreviated either as SST4 or as SSTR4.

Please carefully check the spelling of SST. Somatostatin seems to be abbreviated either as SST or as SS.

Please carefully check whether all abbreviations have been introduced prior first use (e.g. molecular dynamic).

Methods:

ll. 287-298: The authors state that „ The simulation box was filled with TIP3P explicit [81] water molecules and counter ions to neutralize the total charge of the system.“ What was the pH of the fluid within the simulation box? What was the temperature of the fluid?

Discussion:

It would be helpful if the authors could discuss the potential biological consequences of the observed differences between SSTR2-SST and SSTR4-SST binding.

Conflict of interest:

Two of the authors are employed by pharmaceutical companies (Algonist GmBH Austria and PharmInVitro Ltd, Hungary) that specialize in drug discovery for (among others) chronic pain. In the conclusion, the authors state that their results will be useful in future drug design projects. Please make sure that any conflicts of interests can be ruled out.

Author Response

Reviewer 2

The manuscript entitled “Exploration of somatostatin binding mechanism to somatostatin receptor subtype 4“ by Börzsei et al. investigates the chemical basics of endogenous ligand binding to SSTR4. The study was conducted by an in silico approach using BLAST search to find the adequate model of SSTR4 and investigate its interaction with SST. The study seems to be appropriately designed and carefully performed. There are some points that I would like to address:

Introduction:

  1. 46-48: “As a neurotransmitter SST plays role in lots of mechanisms centrally like pain transmission, motor and mood coordination, learning and behavioral responses to stress, locomotor activity [3,10–13]“. I have not yet come across a study showing that SST influences motor coordination and/or locomotor activity. To my knowledge, there is no original article showing that SST influences locomotor activity and motor coordination (motor coordination and locomotor activity does not refer to GI tract motility). The articles cited by the authors do not support that statement.

The Reviewer is acknowledged for his/her careful work and suggestions. In agreement with the Reviewer’s remark, the text on SST influence on motor coordination and/or locomotor activity was deleted.

  1. 69-71: Please clarify the first part of this sentence, the statement of the first main clause is not entirely clear. “SSTR4 mediates anti-inflammatory and anti-nociceptive actions of SST without any endocrine effects releasing from the capsaicin sensitive sensory nerve endings at the periphery [38], and centrally it is involved in learning and memory processes [10] and anxiety and depression-like behavior [37].

We agree that the above cited sentence was somewhat confusing. The following sentence is inserted into the text (lines: 69-71), instead of the first main clause: „SST elicits anti-inflammatory and anti-nociceptive actions and can be released from the capsaicin-sensitive sensory nerve endings. This is mediated through the activation of SSTR4.”

Please carefully check the spelling of SSTR4. Somatostatin receptor 4 for example seems to be abbreviated either as SST4 or as SSTR4.

Please carefully check the spelling of SST. Somatostatin seems to be abbreviated either as SST or as SS.

Please carefully check whether all abbreviations have been introduced prior first use (e.g. molecular dynamic).

SST4 and SS are changed to the correct SSTR4 and SST nomenclature, the changes are highlighted by green background.

Abbreviation of molecular dynamics (MD) is now introduced in the abstract and highlighted with green background.

Methods:

  1. 287-298: The authors state that „The simulation box was filled with TIP3P explicit [81] water molecules and counter ions to neutralize the total charge of the system.“ What was the pH of the fluid within the simulation box? What was the temperature of the fluid?

In the Methods section, in lines 340-342, the temperature of the simulations is indicated (310.15K), and a new line is inserted with green background: “The protonation states of amino acids were set according to pH 7.4.”

Discussion:

It would be helpful if the authors could discuss the potential biological consequences of the observed differences between SSTR2-SST and SSTR4-SST binding.

The following sentence is inserted into the text (lines: 403-406): “In good agreement with experimental results, we found that the extracellular regions of helices and loops play an important role in SST binding and structural differences in these regions are important in receptor subtype selectivity. The detailed structural comparison of SST binding to SSTR2 and 4 helps the development of new, subtype and disease-selective SST analogues.”

Conflict of interest:

Two of the authors are employed by pharmaceutical companies (Algonist GmBH Austria and PharmInVitro Ltd, Hungary) that specialize in drug discovery for (among others) chronic pain. In the conclusion, the authors state that their results will be useful in future drug design projects. Please make sure that any conflicts of interests can be ruled out.

In agreement with the Reviewer’s suggestion, the following text was inserted in the manuscript. „Conflicts of Interest: E. Pinter and Zs. Helyes are co-founders and shareholders of Algonist GmBH Austria and PharmInVitro Ltd, Hungary focusing on drug discovery for (among others) chronic pain and services in pain models, respectively. They declare no conflict of interests with the present work.”